# Comparison Table Generation from Knowledge Bases

Arnaud Giacometti, Béatrice Markhoff, and Arnaud Soulet

Université de Tours, LIFAT, Blois, France
`firstname.lastname@univ-tours.fr`

**Abstract.** Comparison table is an efficient tool for comparing a small number of entities for decision making to analyze the main similarities and differences. The manual choice of their comparison features remains a complex and tedious task. This paper presents VERSUS, which is the first automatic method for generating comparison tables from knowledge bases of the Semantic Web. For this purpose, we introduce the contextual reference level to evaluate whether a feature is relevant to compare a set of entities. This measure relies on contexts that are sets of entities similar to the compared entities. Its principle is to favor the features whose values for the compared entities are reference (or frequent) in these contexts. We show how to select these contexts and how to efficiently evaluate the contextual reference level from a public SPARQL endpoint limited by a fair-use policy. Using our publicly available benchmark based on Wikidata, the experiments show the interest of the contextual reference level for identifying the features deemed relevant by users with high precision and recall. In addition, the proposed optimizations significantly reduce the execution time and the number of required queries.

## 1 Introduction

A comparison table is a double-entry table with entities that you want to compare in columns and comparison features in rows. The comparison table is a particularly useful tool for decision making by isolating the common points and major differences between compared entities. Therefore, this analytical technique is popular in science to compare works, in culture to compare art works or in commerce to compare products or services. This paper aims to fully automate the process of generating a comparison table of a set of entities by querying a knowledge base available on the Semantic Web such as DBpedia [2], YAGO [13] or Wikidata [16]. For instance, starting from Ada Lovelace and Alan Turing, we want to obtain a comparison table like the one presented by Table 1 built from Wikidata (the last column related to our method will be explained later). Beyond people, we aim to compare any entities such as places (countries, cities), objects (tapestries, statues), institutions (universities, political parties), events (tournaments, festivals) and so on. Unfortunately, there is no theoretical framework for the design of comparison tables to determine if a feature is interesting for comparing entities. This task is non-trivial since in 17% of the cases a human

evaluator does not know whether a feature is interesting or not (see Section 6.1 for details). In Table 1, it seems natural to use gender to compare two people. Besides, specifying that Turing was a member of the Royal Society is only interesting because two scientists are compared. Thus, the main challenge is to formalize the notion of *interesting* comparison feature. In addition, we want to benefit from the huge knowledge bases available on the Semantic Web, which raise a problem of robustness and efficiency. Indeed, these knowledge bases are relatively reliable but they most often suffer from incompleteness [10,18]. For this reason, it would be desirable that a feature considered interesting at a given moment remains so despite the subsequent addition of facts. For instance, in Table 1, completing Ada Lovelace's religion should not prevent the selection of "religion" as a comparison feature. Furthermore, rather than centralizing the data, we would like to directly query public SPARQL endpoints to build the comparison tables. This has the advantage of guaranteeing an optimal level of freshness. Nevertheless, the fair-use policy of these public endpoints, which cut off queries that are too expensive, raises optimization needs [12].

| Features | Ada Lovelace | Alan Turing | $crl$ |
|---:|---|---|---|
| sex or gender | female | male | 0.908 |
| spoken language | English | English | 0.472 |
| member of | | Royal Society | 0.205 |
| field of work | mathematics, computing | mathematics, logic, cryptanalysis, cryptography, computer science | 0.110 |
| manner of death | natural causes | suicide | 0.100 |
| religion | ? | atheism | 0.015 |

**Table 1.** A comparison table of Ada Lovelace and Alan Turing as running example

Along these lines, we investigate the first fully-automatic method for generating comparison tables for a particular set of entities without any information other than the knowledge base (meaning no manually-specified comparison features). Our entity-centric approach leads to the following contributions:

- We define a new interestingness measure, called *contextual reference level*, in order to judge if a feature is relevant for comparing a given set of entities. Its principle is to favor the reference features whose values are often used by other sets of similar entities, called *contexts*.
- We show with VERSUS how to select the contexts and how to efficiently evaluate the contextual reference level of a feature while minimizing the number of knowledge base queries. The idea is to estimate bounds and to interrupt the computation as soon as its interest is guaranteed or not.
- We evaluate VERSUS on a publicly available benchmark, named *Comparison Feature Benchmark* (CFB), that we developed to assess the quality of comparison features. It relies on comparison tables built from Wikidata and manually evaluated. On this benchmark, the contextual reference level leads, with equal precision, to better recall and better accuracy than the

state-of-the-art baselines including automatic facet generation. In addition, our optimized evaluation is significantly faster as it requires fewer queries.

The rest of the paper is organized as follows. Section 2 presents related work. Section 3 formalizes our problem. We introduce the contextual reference level and the Versus algorithm in Sections 4 and 5. The experiments in Section 6 evaluate the approach qualitatively and quantitatively.

## 2   Related Work

To the best of our knowledge, there is no work to build a comparison table of a set of entities. We could resort to machine learning methods that learn to rank RDF properties [4]. Unfortunately, it would be difficult to gather feedback specific to our problem to build a training dataset. Besides, as the ranking of the properties depends on the compared entities (for example, `located in` is relevant for only 84.3% of comparison tables in our benchmark), this would require the construction of a training dataset of considerable size to cover all cases.

Most techniques that compare two entities in a knowledge base rely on a similarity measure [1,3]. Such measures are relevant for estimating the resemblance between two entities, but they do not explicitly give the comparison features [14]. In this direction, [9] builds relation paths in knowledge bases between two entities to identify all similarities and differences. Unfortunately, no interestingness measure filters out irrelevant paths leading to too many attributes (including irrelevant ones like identifiers). The tasks closest to ours are the infobox template generation [11,17] and the facet extraction [5,6,8,15]. First, an infobox is a set of attribute-value pairs describing an entity. The choice of attributes is based on a template defined for each class (grouping a set of entities). For instance, persons[1] are described by their name, birth date, nationality, highlights and so on. Many templates have been produced collaboratively by Wikipedia contributors, but methods have also been proposed to automatically refine these templates for more specific classes [11,17]. More recently, [11] proposed an unsupervised metric-based method that favors frequent attributes with popular values with respect to the PageRank. Unfortunately, as for [9], most of the attributes of the infoboxes describe *one* entity in a singular way and therefore cannot be used to compare *several* entities. For instance, image or notable works are not features that can be shared by two persons.

Second, faceted search consists of restricting a collection of entities by selecting only those with a certain value for a given attribute, called facet [15]. A relevant facet has frequently shared values among the observed entities. Typically, facets are temporal (publication date, birth date), spatial (conference location, birth place), personal (author, friend), material (subject, color) or energetic (activity, action) attributes [15]. There are a few automatic facet extraction methods. For a given class, [6] extracts from the infobox templates the attributes whose values are frequently observed. Similarly, [8] measures the quality of an

---

[1] `https://en.wikipedia.org/wiki/Template:Infobox_person`

attribute by favoring frequently used attributes whose values are few and uniformly distributed. Recently, [5] proposes very similar measures to extract facets but a preprocessing method groups the quantitative values (which we do not consider in this paper) and a postprocessing method filters out the redundant facets. These methods mainly derive attributes for a limited number of classes containing a lot of entities. Evaluating very similar entities (like Ada Lovelace and Alan Turing) requires considering smaller groups of much more specific entities (e.g., persons employed by the University of Cambridge). Therefore, the main limitation of automatic facet extraction methods is to miss some very specific but very relevant features. Finally, unlike the facets used for navigation, it does not matter if a comparison feature has a lot of values in the knowledge base with an unbalanced distribution.

## 3    Problem Statement

A knowledge base on a set of relations $\mathcal{R}$ and a set of constants $\mathcal{E}$ (representing entities and values) is a set of facts $\mathcal{K} \subseteq \mathcal{R} \times \mathcal{E} \times \mathcal{E}$. We write the facts in the form $r(s, o) \in \mathcal{K}$, where $r$ is the relation, $s$ is the subject and $o$ is the object. For instance, `religion`(`Turing`, `atheism`) indicates that Alan Turing was an atheist[2]. Given a relation $r$, $r^{-1}(s, o) \in \mathcal{K}$ means that $r(o, s) \in \mathcal{K}$ where $r^{-1}$ is the inverse relation of $r$. Besides, $r_{\mathcal{K}}(s)$ (or more simply, $r(s)$ when the knowledge base $\mathcal{K}$ is clear) is the set of objects associated to the subject $s$ for the relation $r$ in $\mathcal{K}$. For instance, `field of work`(`Turing`) returns the set {`mathematics`, `logic`, `computer science`, `cryptanalysis`, `cryptography`}.

The notion of comparison table is formalized as follows:

**Definition 1 (Comparison table).** *Given a knowledge base $\mathcal{K}$, the comparison table of a set of entities $E \subseteq \mathcal{E}$ by a set of features $F \subseteq \mathcal{R}$ is a table with $|F|$ rows and $|E|$ columns where each cell intersecting a feature $f$ and an entity $e$ contains the values $f(e) = \{o \in \mathcal{E} : f(e, o) \in \mathcal{K}\}$.*

Definition 1 limits the comparison features to the relations of the compared entities. With this definition, to use relation paths [5,9] of greater length (such as "the country of the birth place"), it would be necessary to enrich the knowledge base (which we do not consider in this paper). Table 1 illustrates Definition 1 with the comparison table of the set of entities $E = \{$`Lovelace`, `Turing`$\}$ by the set of features $F = \{$`sex or gender`, `spoken language`, ...$\}$. The cell at the intersection of `field of work` and `Turing` contains the values `field of work`(`Turing`).

An interestingness measure $m : \mathcal{R} \times 2^{\mathcal{E}} \times 2^{(\mathcal{R} \times \mathcal{E} \times \mathcal{E})} \to [0, 1]$ evaluates the interest $m(f, E, \mathcal{K})$ of using the relation $f$ as a feature for comparing the entities of $E$ in the knowledge base $\mathcal{K}$.

**Definition 2 (Interesting feature).** *Given a KB $\mathcal{K}$, a set of entities $E \subseteq \mathcal{E}$, an interestingness measure $m : \mathcal{R} \times 2^{\mathcal{E}} \times 2^{(\mathcal{R} \times \mathcal{E} \times \mathcal{E})} \to [0, 1]$ and a threshold $\gamma \in [0, 1]$, an interesting feature $f \in \mathcal{R}$ (for $m$ and $\gamma$) satisfies $m(f, E, \mathcal{K}) \geq \gamma$.*

---

[2] The `Typewriter` font denotes the literals from Wikidata that are used as illustrations.

**Given a KB $\mathcal{K}$, a set of entities $E$, an interestingness measure $m$ and a threshold $\gamma$, we aim at extracting all the interesting features $F = \{f \in \mathcal{R} : m(f, E, \mathcal{K}) \geq \gamma\}$ to build a comparison table of $E$ by $F$.**

For this purpose, we have to address two challenges. The first challenge consists in defining an interestingness measure that estimates the relevance of a feature from a knowledge base (see Section 4). The second challenge is to efficiently evaluate this measure by minimizing the number of SPARQL queries (see Section 5).

## 4    Contextual Reference Level of a Feature

### 4.1    Definition

Intuitively, to understand and to be able to interpret a comparison table, a feature is interesting if the values describing the compared entities are known by the user. In psychology, it is well known that the user needs at least one *reference* value to compare two values [14]. In particular, if these values are too rare (or even only characterize one compared entity), the user of the table is unlikely to know them because he has never been confronted with them. Sometimes such values are informative, but they do not help to compare the entities with each other. For instance, the place of burial of Ada Lovelace is Hucknall Church St Mary Magdalene while that of Alan Turing is Woking Crematorium. There is no particular conclusion to draw from this difference (except perhaps that Alan Turing was atheist unlike Ada Lovelace, but the feature `religion` is much better suited to underline this point). Of course, this notion of scarcity is dependent on the compared entities. Even if there are only few people who are members of the Royal Society, this feature makes sense to compare two persons employed by the University of Cambridge. The key idea of our interestingness measure is to evaluate the relevance of a feature according to entities that are similar to the compared entities (for instance, those "being employed by Cambridge" or those "speaking English" for Ada Lovelace and Alan Turing). We formalize this intuition by introducing the notion of context:

**Definition 3 (Context).** *Given a set of entities $E \subseteq \mathcal{E}$ and a relation-object couple $(r, o) \in \mathcal{R} \times \mathcal{E}$ such that $E \subseteq r^{-1}(o)$, the context $C$ for $E$ stemming from $(r, o)$ is the set of entities $r^{-1}(o) \backslash E$. $\mathbb{C}_E$ denotes the set of all contexts for $E$.*

Intuitively, a context $C$ is a set of entities that are similar but different from the entities of $E$ with respect to a relation-object couple $(r, o)$ shared by all the entities of $E$. For the comparison table provided by Table 1, an example of context is the set of entities having English as spoken language (here, the relation-object couple is (`spoken language`, `English`)). Naturally, the classes are conducive to contexts. For example, all persons (i.e., entities with couple (`instance of`, `human`)) could constitute a context for Lovelace and Turing.

Given a set of entities $E \subseteq \mathcal{E}$, a feature $f \in \mathcal{R}$ and a context $C \in \mathbb{C}_E$, the more the set of values $f(e)$ with an entity $e \in E$ describes the entities of $C$, the

more this feature $f$ has a chance to be a referent for the user of the table. From this intuition, given an entity $e \in E$, we deduce that the interest of a feature $f$ should increase with the probability of observing the values in $f(e)$ in the set of values $f(s)$ of similar entities $s \in C$: $\Pr\big[f(s) \cap f(e) \neq \emptyset \mid s \in C\big]$. Then, given a set of entities $E = \{e_1, \ldots, e_P\}$ and a context $C \in \mathbb{C}_E$, we define the *contextual reference level* of a feature $f$, denoted by $crl_C(f, E, \mathcal{K})$, as the probability of observing the values $f(e_i)$ of at least one entity $e_i \in E$ in the set of values $f(s_i)$ of similar entities $s_i \in C$:

$$crl_C(f, E, \mathcal{K}) = \Pr\big[(f(s_1) \cap f(e_1) \neq \emptyset) \vee \ldots \vee (f(s_P) \cap f(e_P) \neq \emptyset) \mid s_1 \in C, \ldots, s_P \in C\big]$$
$$= \Pr\big[(\exists e_i \in E)(f(s_i) \cap f(e_i) \neq \emptyset) \mid s_i \in C\big]$$

It is indeed a probability because if a similar entity $s_i$ shares features with several entities in $E$, it is counted only once. In practice, entities belong to several relevant contexts. For example, for Ada Lovelace and Alan Turing, we will consider the contexts stemming from four couples (see Section 5.2 for details): (`field of work`, `mathematics`), (`employer`, `Univ. of Cambridge`), (`occupation`, `computer scientist`) and (`spoken language`, `English`). For this reason, we extend the definition of $crl_C(f, E, \mathcal{K})$ to a set of contexts $\mathcal{C} = \{C_1, \ldots, C_K\}$ as follows:

**Definition 4 (Contextual reference level).** *Given a set of entities $E = \{e_1, \ldots, e_P\} \subseteq \mathcal{E}$ and a set of contexts $\mathcal{C} = \{C_1, \ldots, C_K\} \subseteq \mathbb{C}_E$, the contextual reference level of a feature $f$ is defined as :*

$$crl_{\mathcal{C}}(f, E, \mathcal{K}) = \Pr\Big[(\exists e_i \in E)(\exists k \in [1..K])(f(s_i^k) \cap f(e_i) \neq \emptyset) \mid s_i^k \in C_k\Big]$$

It is important to note that the compared entities $E$ play a very strong role in this definition because they limit the choice of $\mathcal{C}$ in the set of potential contexts $\mathbb{C}_E$. The fourth column of Table 1 indicates the contextual reference level of each feature computed from Wikidata in the four contexts mentioned above. For instance, 0.908 corresponds to the probability of observing the value `female` or `male` (respectively stemming from Ada Lovelace or Alan Turing for $e_i$) as `sex or gender` of an entity $s_i$ that is a mathematician or an employee of Cambridge or a computer scientist or an English speaker. With Definition 4, it would be possible to directly calculate the contextual reference level of a feature with a SPARQL query (not reported here due to the space limit). However, this statistical query would often be too costly not to be interrupted by the fair-use policy of public SPARQL endpoints [12]. Nevertheless, this definition implicitly assumes that the entities $s_i^k$ are identically and independently drawn in the different contexts $C_k$. With this i.i.d. assumption, the following property rewrites the contextual reference level:

*Property 1.* Given a set of entities $E \subseteq \mathcal{E}$, a set of contexts $\mathcal{C} \subseteq \mathbb{C}_E$ and a feature $f \in \mathcal{R}$, we have:

$$crl_{\mathcal{C}}(f, E, \mathcal{K}) = 1 - \prod_{C \in \mathcal{C}} \prod_{e \in E} (1 - \Pr\big[f(s) \cap f(e) \neq \emptyset \mid s \in C\big]) = 1 - \prod_{C \in \mathcal{C}} (1 - crl_C(f, E, \mathcal{K}))$$

Due to lack of space, we omit the proofs, but this follows from Morgan's law. Interestingly, each probability $\Pr\big[f(s) \cap f(e) \neq \emptyset \mid s \in C\big]$ can easily be calculated independently by a low-cost SPARQL query. We will see in Section 6.3 that in practice, the error rate of this kind of query is under 0.5%. In addition, considering Property 1, it is easy to see that the contextual reference level increases with the probability $\Pr\big[f(s) \cap f(e) \neq \emptyset \mid s \in C\big]$ and that its range is $[0, 1]$. The contextual reference level of the feature $f$ is zero when no entity among those of the contexts $\mathcal{C}$ has a common value with the entities of $E$. Conversely, $crl_{\mathcal{C}}(f, E, \mathcal{K})$ is equal to 1 as soon as a value in $f(e)$ is shared by all the entities of at least one context $C$.

### 4.2 Quality criteria analysis

Properties 2-4 present three quality criteria that a well-behaved interestingness measure for evaluating features should satisfy. First, the following property proves that contextual reference level is monotone with respect to contexts:

*Property 2.* Given a KB $\mathcal{K}$, a feature $f$ and a set of entities $E$, we have $crl_{\mathcal{C}}(f, E, \mathcal{K}) \leq crl_{\mathcal{C}'}(f, E, \mathcal{K})$ if the two sets of contexts satisfy $\mathcal{C} \subseteq \mathcal{C}' \subseteq \mathbb{C}_E$.

This result is explained by the addition of factors less than 1 in the double product of Property 1 when a context is added to $\mathcal{C}$. Interestingly, the addition of a new context favors the emergence of new interesting features (e.g., if a new relation is added to the knowledge base). However, we will see in Section 5.2 that this also raises problems of redundancy between contexts. The following property goes further by showing that contextual reference level is also robust against incompleteness for the feature $f$:

*Property 3.* Given two KBs $\mathcal{K}$ and $\mathcal{K}'$, a set of contexts $\mathcal{C} \subseteq \mathbb{C}_E$ and a feature $f$ such that $f_{\mathcal{K}}(e) \subseteq f_{\mathcal{K}'}(e)$ for each $e \in E$, we have $crl_{\mathcal{C}}(f, E, \mathcal{K}) \leq crl_{\mathcal{C}}(f, E, \mathcal{K}')$.

This property underlines that the value of contextual reference level is always underestimated when some facts are missing. If new facts are added in the knowledge base, then the contextual reference level of a feature can only increase (if the context $\mathcal{C}$ remains unchanged). For this reason, the extracted features will remain interesting for $crl$ if the knowledge base is completed. In Table 1, the feature `religion` was selected despite the lack of value for Ada Lovelace. Whatever the value could be stated, this feature would remain interesting for $crl$.

Finally, the next property proves that contextual reference level of a feature $f$ is zero when it is an identifier (i.e., an injective function $f(x) = f(y) \Rightarrow x = y$):

*Property 4.* Given a set of entities $E$ and a set of contexts $\mathcal{C}$, we have $crl_{\mathcal{C}}(f, E, \mathcal{K}) = 0$ for any feature $f$ that is an identifier.

This result is explained by observing that for an injective function $f$, we have $f(e) \cap f(s) = \emptyset$ for any entity $e \in E$ and subject $s \in C$ because the set of entities $E$ and any context $C$ in $\mathcal{C}$ are disjoint (see Definition 3). Interestingly, an identifier $f$ is not relevant for a comparison table because all values of $f$ uniquely identifies an entity. For instance, for a set of countries, the property `GeoNames ID` is not an interesting feature w.r.t. $crl$.

---

**Algorithm 1** Versus: extracting the set of interesting features w.r.t. $crl$

---

**Input:** A knowledge base $\mathcal{K}$, a set of entities $E \subseteq \mathcal{E}$ and a threshold $\gamma$
**Output:** The set of interesting features $F \subseteq \mathcal{R}$
 1: $F := \emptyset$
 2: $\mathcal{R}_E := \{r \in \mathcal{R} : e \in E \wedge r(e, s) \in \mathcal{K}\}$
 3: **for all** $f \in \mathcal{R}_E$ **do**
 4:    Select the set of contexts $\mathcal{C}$ for the entities $E$ and the feature $f$ with Algorithm 2
 5:    **if** $crl_{\mathcal{C}}(f, E, \mathcal{K}) \geq \gamma$ (using Algorithm 3) **then** $F := F \cup \{f\}$
 6: **end for**
 7: **return** $F$

---

## 5   Versus: A Method for Extracting Interesting Features

### 5.1   Overview

The overall idea is to analyze each relation $f$ that describes at least one entity in $E$ to determine whether it is an interesting feature in $\mathcal{K}$: $crl_{\mathcal{C}}(f, E, \mathcal{K}) \geq \gamma$. Algorithm 1 sketches this process. First, the set $F$ that will contain all the interesting features is initialized with the empty set (Line 1) and the set of all the candidate relations $\mathcal{R}_E$ gathers the relations that describe at least one entity in $E$ (Line 2). After, each relation in $\mathcal{R}_E$ is separately processed (Lines 3-6). Line 4 selects the set of contexts $\mathcal{C} \subseteq \mathbb{C}_E$ without considering the relation $f$ (see Algorithm 2). This set of contexts is immediately used by Algorithm 3 in order to decide whether the relation $f$ is an interesting feature for the entities in $E$. If $f$ is really interesting for $crl$, it is added to the set of interesting features $F$. Finally, this set is returned at Line 7.

The rest of this section details Lines 4 and 5 based respectively on Algorithms 2 and 3. Section 5.2 gives the method for selecting the set of contexts. Of course, this choice is decisive in the calculation of the contextual reference level. Section 5.3 presents an efficient algorithm for evaluating the contextual reference level. Indeed, the naive evaluation of the contextual reference level is expensive, as for each feature, it requires to calculate $|\mathcal{C} \times E|$ queries for the numerators and $|\mathcal{C}|$ queries for the denominators (see Definition 4).

### 5.2   Context selection

This step aims to select a small number of relevant contexts among all the contexts of $\mathbb{C}_E$ that may be redundant. Indeed, in the case where a large number of contexts in $\mathcal{C}$ are correlated, the contextual reference level might be abnormally overestimated (see Property 2). For example, since all employees of the University of Cambridge are necessarily humans, the context stemming from (`instance of`, `human`) does not provide additional information, but it increases the contextual reference level. It is however important to keep a set of contexts that cover all the specificities of the entities similar to $E$: $\bigcap \mathbb{C}_E$. For example, the context stemming from (`occupation`, `computer scientist`) is important because

---

**Algorithm 2** Selecting a set of contexts

---

**Input:** A knowledge base $\mathcal{K}$, a set of entities $E \subseteq \mathcal{E}$ and a feature $f \in \mathcal{R}$
**Output:** A set of contexts $\mathcal{C} \subseteq \mathbb{C}_E$
1: $\mathcal{C} := \{r^{-1}(o) \backslash E : r \in (\mathcal{R} \backslash \{f\}) \wedge (\forall e \in E)(r(e, o) \in \mathcal{K})\}$
2: Sort the contexts of $\mathcal{C}$ by ascending cardinality
3: **for all** context $C_i \in \langle C_1, \ldots, C_n \rangle$ **do**
4:    **if** $\bigcap (\mathcal{C} \setminus C_i) = \bigcap \mathcal{C}$ **then** $\mathcal{C} := \mathcal{C} \setminus C_i$
5: **end for**
6: **return** $\mathcal{C}$

---

| Relation $r$ | Object $o$ | $\|r^{-1}(o) \backslash E\|$ |
|---:|:---|---:|
| field of work | mathematics | 2,018 |
| employer | Univ. of Cambridge | 3,129 |
| occupation | computer scientist | 7,943 |
| ~~described by source~~ | ~~Obalky knih.cz~~ | 47,563 |
| spoken language | English | 165,714 |
| ~~instance of~~ | ~~human~~ | 6,389,426 |

**Table 2.** Relation-object couples common to Ada Lovelace and Alan Turing

it distinguishes Ada Lovelace and Alan Turing from mathematicians at the University of Cambridge who have not contributed in computer science. In this way, we choose one of the smallest sets of contexts $\mathcal{C}^* \subseteq \mathbb{C}_E$ that characterizes the same set of entities as $\mathbb{C}_E$ by intersecting: $\mathcal{C}^* \in \arg\min_{\mathcal{C} \subseteq \mathbb{C}_E}\{|\mathcal{C}| : \bigcap \mathcal{C} = \bigcap \mathbb{C}_E\}$. The exact resolution of this problem is NP-hard and it would require a large number of knowledge base queries. We therefore propose a heuristic algorithm, which eliminates superfluous contexts from the smallest one to the largest one.

Given a knowledge base $\mathcal{K}$, a set of entities $E$ and a feature $f$, Algorithm 2 returns a set of contexts $\mathcal{C}$. Line 1 builds the set of contexts $\mathbb{C}_E$ except it excludes the context stemming from the feature $f$ (i.e., $r \neq f$). The contexts are then sorted from the smallest to the largest (Line 2) to favor the removal of overly general contexts. The loop (Lines 3-5) iterates over each context $C_i$ starting with the smallest one. Line 4 tests whether the intersection of contexts without $C_i$ gives the same set of entities as with $C_i$. If this is the case, it means that this context does not provide any specificity and it is discarded from $\mathcal{C}$. Once the loop is completed, Line 6 returns the set of non-redundant contexts.

Table 2 presents the relation-object couples $(r, o)$ from which contexts are computed considering Ada Lovelace and Alan Turing. After having been sorted by ascending cardinality in Wikidata (i.e., $|r^{-1}(o) \backslash E|$), the two redundant contexts were eliminated by Lines 3-7 of Algorithm 2. For example, the restriction "instance of human" does not delete any entity among those belonging to all other contexts. It is important to note that the interest of an approach centered on entities is to consider contexts that do not depend only on classes (i.e., there are other relations than `instance of`). However, the number of contexts in $\mathcal{C}$ remains reasonable in practice (7 at most in our experiments). Most often, the

iteration of Lines 3-5 removes few contexts, but in some cases, many redundant contexts are eliminated (for example, 167 in the most extreme case).

### 5.3   Efficient evaluation of the contextual reference level

Rather than calculating the exact contextual reference level of a feature, the idea is to do a partial calculation of this value in order to only determine whether $crl_{\mathcal{C}}(f, E, \mathcal{K})$ is greater than $\gamma$. It is easy to see that the complement to 1 of the contextual reference level (i.e., $1 - crl_{\mathcal{C}}(f, E, \mathcal{K})$) decreases with each multiplication by a factor of the form $\left(1 - \Pr\left[f(s) \cap f(e) \neq \emptyset \mid s \in C\right]\right)$. With this observation, it is possible to derive a lower bound for the contextual reference level. In the process of calculation, when this lower bound exceeds the threshold $\gamma$, we have the guarantee that $crl_{\mathcal{C}}(f, E, \mathcal{K}) \geq \gamma$. Conversely, it is possible to derive an upper bound of the contextual reference level by using $\Pr[f(s) \cap f(e) \neq \emptyset, s \in \mathcal{E}]$ as an upper bound of the joint probability $\Pr[f(s) \cap f(e) \neq \emptyset, s \in C]$. The following property formalizes these two bounds:

*Property 5.* Given a knowledge base $\mathcal{K}$, the contextual reference level of a feature $f$ for the entities $E$ is bounded for any $\mathcal{S} \subseteq \mathcal{C} \times E$:

$$crl_{\mathcal{C}}(f, E, \mathcal{K}) \geq 1 - \prod_{(C,e) \in \mathcal{S}} \left(1 - \Pr\left[f(s) \cap f(e) \neq \emptyset \mid s \in C\right]\right)$$

$$\leq 1 - \left[ \prod_{(C,e) \in \mathcal{S}} \left(1 - \Pr\left[f(s) \cap f(e) \neq \emptyset \mid s \in C\right]\right) \right.$$

$$\left. \times \underbrace{\prod_{(C,e) \in (\mathcal{C} \times E) \setminus \mathcal{S}} \left(1 - \frac{\min\{|\{s \in \mathcal{E} : f(s) \cap f(e) \neq \emptyset\}|, |C|\}}{|C|}\right)}_{\text{optimistic factor}} \right]$$

Algorithm 3 benefits from these bounds for efficiently evaluating if $crl_{\mathcal{C}}(f, E, \mathcal{K}) \geq \gamma$. More precisely, Lines 1 and 2 respectively initialize the product $p$ and the optimistic factor $o$ discussed above by considering all the couples in $\mathcal{C} \times E$. The loops of Lines 3 and 4 enumerate the different entities $e \in E$ and the different contexts $C \in \mathcal{C}$. At each iteration, Line 5 refines the calculation of $p$ taking into account the probability $\Pr\left[f(s) \cap f(e) \neq \emptyset \mid s \in C\right]$ while Line 7 updates $o$. If the current contextual reference level is higher than the threshold $\gamma$, Line 6 returns *true* because $1 - p$ is a pessimistic approximation of the final contextual reference level. Conversely, Line 8 returns *false* when the upper bound $1 - p \times o$ is lower than $\gamma$.

Let us illustrate Algorithm 3 with the computation of the contextual reference level of two features for Ada Lovelace and Alan Turing with a threshold $\gamma = 0.01$ illustrated by Table 3. The probability of having an entity with a natural death (like Ada Lovelace) among those who studied mathematics is 0.025. From this evaluation, it is certain that `manner of death` is an interesting feature because its exact contextual reference level exceeds the lower bound $1 - (1 - 0.025)$ which is higher than the threshold $\gamma$. In this case, this avoids the evaluation of 7

---

**Algorithm 3** Computing the contextual reference level of a relation

---

**Input:** A knowledge base $\mathcal{K}$, a set of entities $E \subseteq \mathcal{E}$, a threshold $\gamma$, a set of contexts $\mathcal{C}$ and a relation $f$

**Output:** Return true if the relation $f$ is interesting i.e., $crl_{\mathcal{C}}(f, E, \mathcal{K}) \geq \gamma$

1: $p := 1$
2: $o := \prod_{(C,e) \in \mathcal{C} \times E} \left( 1 - \frac{\min\{|\{s \in \mathcal{E} : f(s) \cap f(e) \neq \emptyset\}|, |C|\}}{|C|} \right)$
3: **for all** $e \in E$ **do**
4:     **for all** $C \in \mathcal{C}$ **do**
5:         $p := p \times (1 - (|\{s \in C : f(s) \cap f(e) \neq \emptyset\}|)/(|C|))$
6:         **if** $1 - p \geq \gamma$ **then** $true$
7:         $o := o / \left( 1 - \frac{\min\{|\{s \in \mathcal{E} : f(s) \cap f(e) \neq \emptyset\}|, |C|\}}{|C|} \right)$
8:         **if** $1 - p \times o < \gamma$ **then** $false$
9:     **end for**
10: **end for**
11: **return** $false$

---

|  | | Contexts $\mathcal{C}$ | | | |
|---|---|---|---|---|---|
|  | Entities $E$ | field of work | employer | occupation | spoken language |
| manner of death | Lovelace | **0.025** | 0.018 | 0.017 | 0.038 |
|  | Turing | 0.002 | 0.001 | 0.002 | 0.003 |
| student of | Lovelace | **0,000** | 0,000 | 0,000 | 0,001 |
|  | Turing | 0,001 | 0,000 | 0,000 | 0,001 |

**Table 3.** Computation of *crl* of two features for Ada Lovelace and Alan Turing

queries that would have been necessary for the exact calculation of the contextual reference level. For the feature student of, the optimistic factor after the first evaluation is equal to 0.998. It is therefore sure that the contextual reference level of this feature is at most equal to $1 - (1 - 0) \times 0.998 = 0.002$ which is lower than the threshold $\gamma$. Again, the contextual reference level computation can be interrupted (Line 8) avoiding the evaluation of 7 queries.

## 6  Experiments

After presenting the evaluation benchmark in Section 6.1, our experiments aim to answer the following two questions: Does the contextual reference level really isolate the best features? (Q1) and What is the gain of the optimized evaluation? (Q2). These questions are respectively answered in Sections 6.2 and 6.3. VERSUS is implemented in Java using the Jena library to query the public Wikidata SPARQL endpoint. VERSUS was run on Windows 10 with an Intel core i7 processor and 32 GB of RAM. Due to the few operations performed on the client side, the execution times correspond essentially to the processing time of SPARQL

queries on the server side[3]. Although execution times vary with server load and available data, they shape the behavior of the approaches. Note that the source code of Versus, the evaluation tool and the results are available on the website `https://lovelace-vs-turing.com` and `github.com/asoulet/versus`.

### 6.1   Comparison Feature Benchmark (CFB)

As comparison table generation is a new problem, we had to develop a benchmark, named *Comparison Feature Benchmark* (CFB). Its twofold objective is to constitute a reference dataset to compare the speed of different approaches and a gold standard to assess the quality of the discovered comparison features. This section starts by describing the method to select from Wikidata the sets of entities to be compared with their candidate comparison features. For simplicity, we consider only pairs of entities to be compared. Then, we explain how the quality of the candidate comparison features have been manually evaluated.

First, we randomly draw from Wikidata 1,000 types $T_i$ ($i \in [1..1000]$) that have between 10k and 1k instances. This random sample guarantees to cover a wide variety of entities (person, place, objects, events and so on) in order to best reflect Wikidata diversity. Second, for each type $T_i$, we select the two entities $e_i^1$ and $e_i^2$ that have the highest degree of incoming facts (i.e., maximizing $\deg(e) = |\{s \in \mathcal{E} : r(s, e) \in \mathcal{K} \wedge e \in T_i\}|$). This in-degree ranking favors popular entities of the type $T_i$. For instance, the entities `Paris` (Q90) and `London` (Q84) are selected for the type `city` (Q515). Then, for each pair of entities $E_i = \{e_i^1, e_i^2\}$, we define the set $F_i$ of relations $r_j$ where $r_j \in \mathcal{R}$ has URI as objects, $r_j$ is a direct property of Wikidata (by using the prefix `http://www.wikidata.org/prop/direct/`) and $r_j(e_i^1)$ or $r_j(e_i^2)$ is not empty (note that we do not consider the inverse relations less likely to be features). Thus, $F_i$ is the set of candidate comparison features to compare entities in $E_i$. Finally, for each pair of entities $E_i = \{e_i^1, e_i^2\}$, we store in our benchmark CFB all the facts $r_j(e_i^k, o_i^k)$ ($k \in \{1, 2\}$) where $r_j \in F_i$ and $o_i^k$ is an object randomly drawn from the values in $r_j(e_i^k)$ (if $r_j(e_i^k)$ is the empty set, then $o_i^k$ is null). For each type $T_i$ ($i \in [1..1000]$), this process builds a comparison table with $|F_i|$ rows and two columns to compare $e_i^1$ and $e_i^2$.

Second, 1,195 candidate features (out of the 11,852, or about 10%) were drawn at random and evaluated manually by one of the 6 evaluators. Each time, we asked if the candidate feature $r_j \in F_i$ was relevant to compare the pair of entities $E_i = \{e_i^1, e_i^2\}$ (by selecting in CFB the facts $r_j(e_i^1, o_i^1)$ and $r_j(e_i^2, o_i^2)$). The evaluator can answer "No", meaning that the feature $r_j$ is not relevant (44.9% of the evaluations), "Yes" (37.9%) or "I don't know" (17.2%). Only 80 evaluations were common, of which 74 agreed. It leads to a Cohen's kappa coefficient of 0.832 that corresponds to an *almost perfect* agreement [7].

### 6.2   Q1: Quality of the extracted features

Figures 1 and 2 respectively report the precision-recall results (ignoring "I don't know" evaluations) and the number of comparison features. First, we benefit

---

[3] `https://query.wikidata.org/`

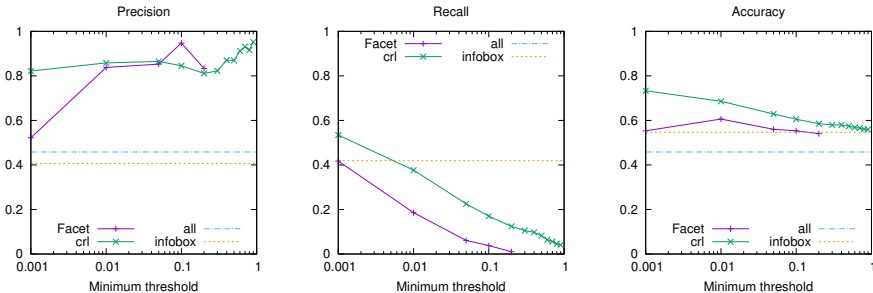

**Fig. 1.** Precision, recall and accuracy for crl, Facet, infobox and all

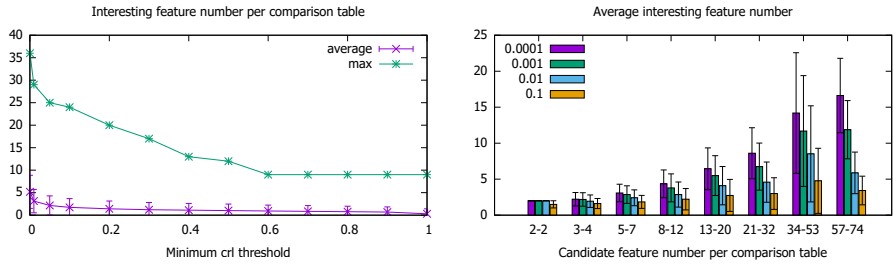

**Fig. 2.** Number of interesting features per comparison table

from the CFB benchmark described in the previous section for comparing the contextual reference level used by Versus (denoted by crl) with the metric used by automatic facet generation [8] (denoted by Facet). For the facet-oriented metric, the type $T_i$ of the two entities (see above) is used to define the collection on which the metric is computed. We also use two baselines: the all method [9] that selects all the candidate features of the benchmark and the infobox method that selects all the candidate features present in at least one of the Wikipedia infoboxes of the entities $E_i$. Figure 1 reports the precision, recall and accuracy for these methods by varying the minimum threshold for crl and Facet. For the reasons mentioned in related work, we observe that the precision of all and infobox, less than 50%, is catastrophic. When the precision of Facet is better than that of crl, the recall of Facet is dramatically low (less than 20 features are extracted). Overall, the contextual reference level is much better than facet-oriented metric with comparable precision but higher recall and higher accuracy. This result is not surprising because, unlike Facet, our method brings out features specific to the two compared entities.

The precision of the contextual reference level, always above 76%, is generally high with regard to the baselines whose precision is less than 50%. Interestingly,

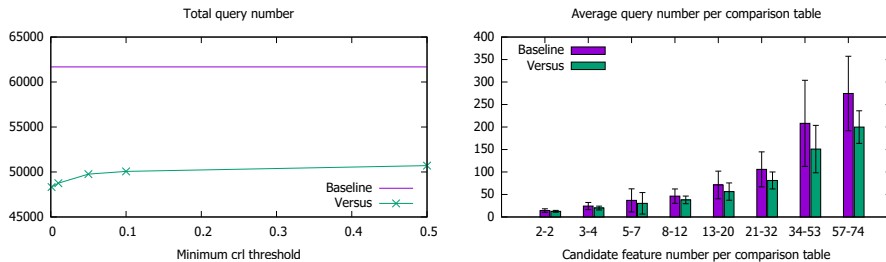

**Fig. 3.** Number of SPARQL queries executed on Wikidata

this precision increases with the minimum contextual reference level threshold (from 76% for $\gamma = 0.0001$ to 86% for $\gamma = 0.05$). This demonstrates the ability of our measure to isolate the most relevant features. However, the recall decreases very quickly with the minimum contextual reference level threshold. This is explained by the decrease in the number of interesting features with $\gamma$ as shown in Figure 2. With $\gamma = 0.1$, the left-hand side graph indicates that a comparison table contains only three features on average. However, the right-hand side graph shows strong disparities depending on the initial number of candidate features describing the entities (note that slices are non-linear). In practice, to have a good compromise, it seems appropriate to set $\gamma$ with a value less than 0.1.

### 6.3  Q2: Efficiency of the method

This section assesses the efficiency gain of the optimized method (VERSUS benefiting from Property 5) with a baseline where the exact value of the contextual reference level is calculated (baseline based on Property 1).

Figure 3 indicates the number of SPARQL queries required to build the 1,000 comparison tables of the benchmark. The left-hand side figure plots the total number of queries required by baseline and by VERSUS with respect to the minimum crl threshold. It is always more advantageous to use the optimized method because fewer queries are executed (around -20% of queries). VERSUS is even more efficient for low thresholds (i.e., $\gamma \leq 0.1$). For $\gamma = 0.01$, we observe on the right-hand side figure that the number of queries increases linearly with the number of candidate features to be tested. This result is expected because the number of contexts (between 1 and 7) is relatively independent of the number of features. Again, VERSUS is always more efficient.

The average execution time for building a comparison table with VERSUS is 58.8 seconds. In the worst case, it requires 693.1 seconds to generate that of two authors, namely the botanist Miguel Colmeiro and the poet Manuel Curros Enríquez. More precisely, Figure 4 (left-hand side) details the average execution time to construct a comparison table according to the number of features. Unlike the number of queries, the execution time of the construction of a comparison

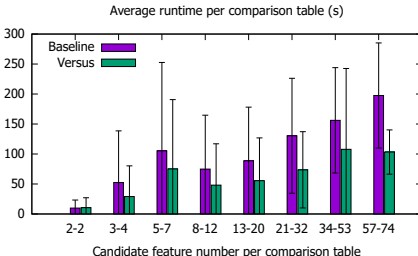
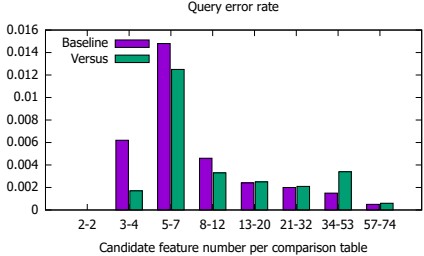

**Fig. 4.** Average running time and query error rate

table does not increase linearly with the number of features. In addition, the standard deviations are very high. This phenomenon is explained by the fact that not all queries have the same complexity. For instance, it is more expensive to evaluate a query with the person as context than with the country as context because the latter contains fewer entities. In particular, very few queries ($< 0.5\%$) have an execution time that exceeds the limit of the fair-use policy of Wikidata and fail as shown by the error rates on Figure 4 (right-hand side). Interestingly, the gain in execution time of VERSUS is around 30%, better than the gain in number of queries (only 20%).

## 7  Conclusion

We presented VERSUS that automatically generates a comparison table of a set of entities from a knowledge base by querying its public SPARQL endpoint. To this end, we introduced the contextual reference level that evaluates whether a feature has values for the compared entities which are sufficiently common among other similar entities. We have broken down the computation of the contextual reference level into several low-cost SPARQL queries so that it satisfies the fair-use policy of Wikidata public endpoint. Finally, this computation is also optimized in VERSUS to reduce this number of queries. Experiments on our Comparison Feature Benchmark show the good precision of the contextual reference level for isolating the most relevant features. Interestingly, our entity-centric approach has a higher recall and accuracy than a baseline using facet-oriented metric, which relies on classes. Moreover, thanks to our optimization, VERSUS is about 30% faster than a naive approach. In future work, we would like to investigate other kinds of interestingness measures not based on the contextual reference level, but on the contrary, on exceptionality. If such measures are likely to have a weak recall, they could be used in addition to the contextual reference level for extracting unexpected features. Instead of evaluating each feature one by one, it would also be relevant to extract an interesting *set* of features so as to

avoid redundancies. This would be essential to combine several endpoints from the Linked Open Data cloud that necessarily contain repeated information.

**Acknowledgments.** We thank the evaluators for the time they took to annotate the features. This work was partially supported by the grant ANR-18-CE38-0009 ("SESAME").

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
