# OpenReview forum: "Comparison Table Generation from Knowledge Bases"
_eswc-conferences.org/ESWC/2021/Conference/Research_Track — ESWC 2021 Research_

### Official Review · AnonReviewer4 · 2021-01-05
**Interesting study, but limited evaluation and comparison with related work**

**Rating:** 1
**Confidence:** 5
**Impact:** 3
**Design And Technical Quality:** 3

**Review:**

The paper targets the problem of comparison table generation for entities in knowledge graphs. The problem seems novel, and I see scope for a valuable contribution with this work. The paper is well-written, and the VERSUS algorithm appears as an interesting technical contribution, efficiently solving the problem of computing frequencies of properties in comparison sets.

At the same time, the evaluation ignores existing benchmarks and baselines, making it at the moment not clear whether the proposed approach is the best for the problem.

**Anonymity:**

Yes, I would like my review to remain anonymous.

**Reuse And Availability:**

4: High

**Strong Points:**

The problem seems novel, and I see scope for a valuable contribution with this work. The paper is well-written, and the VERSUS algorithm appears as an interesting technical contribution, efficiently solving the problem of computing frequencies of properties in comparison sets.

**Subreviewer:**

I submitted this review.

**Weak Points:**

1. The paper requires some leap of faith from the method to the motivation: While the method performs pointwise ranking, and the experiments only evaluate binarized pointwise interestingness, comparison tables seem to be more complex constructs, probably requiring thought about redundancy, diversity, etc. (which are somewhat mentioned in the method, but not captured by the pointwise evaluation). This relates also to the unclear use case for comparison tables - although I can feel for the difficulty in grounding interestingness, an extrinsic use case for the generated tables would be much helpful to clarify and strengthen this point.
2. The experimental evaluation has serious issues by using pointwise, hard-to-ground annotations, and it is not clear to me what redundancy was built into the annotation, and what agreement was. The authors should evaluated their work on existing benchmarks with proper redundancy [1].
3. The evaluation is also limited as it lacks baselines - the proposed contextual reference levels seem to be a generalization of the peer groups used in Recoin [1,2], and as such the paper should compare with that simple, class-based baseline.
4. The leap from a classification problem (interesting or not) to a ranking problem (which are the top-5/top-10/top-20 properties to include in a comparison table) is also unclear in the experiments.

To me, the problem of comparison table generation is interesting, and the VERSUS algorithm alone for computing contextual reference levels appears a relevant contribution. At the same time, I am not yet convinced that the latter is the best solution for the former, and would recommend that the authors put more effort into evaluation on existing benchmarks, and comparison with baselines.

[1] Razniewski et al., Doctoral Advisor or Medical Condition: Towards Entity-specific Rankings of Knowledge Base Properties, ADMA, 2017
[2] Balaraman et al., Recoin: Relative Completeness in Wikidata, Wiki Workshop, 2018

---

> ### Author Rebuttal · Authors · 2021-01-29
>
> Thank you for the comments.
>
> *W1: comparison tables seem to be more complex constructs, probably requiring thought about redundancy, diversity, etc. (which are somewhat mentioned in the method, but not captured by the pointwise evaluation) \
> W2: it is not clear to me what redundancy was built into the annotation \
> W4: The leap from a classification problem (interesting or not) to a ranking problem (which are the top-5/top-10/top-20 properties to include in a comparison table) is also unclear in the experiments.*
>
> The redundancy between the comparison features is only mentioned in future work of the conclusion. Our problem formulation clearly states that the comparison features are extracted separately (interestingness of the property above a threshold) and our method extracts the features separately. We therefore think that it is normal to evaluate each feature separately. This is also why we did not do a top-k evaluation.
>
> *W2: what agreement was*
>
> Concerning the ground truth, there are 1,275 evaluations covering 1,195 features among 11,853 candidate features coming from 1,000 comparison tables. This random coverage of 10% of the candidate features is sufficient for statistical validity. Regarding the Kappa coefficient, there are 74 agreements (24  $yes$/50 $no$) and 6 disagreements (3 $yes/no$ + 3 $no/yes$) leading to $P_{yes}=0.113$, $P_{no}=0.438$ and Kappa=((74/80) - ($P_{yes}$+$P_{no}$)) / (1 - ($P_{yes}$+$P_{no}$))=0.832.
>
> *W2: The authors should evaluated their work on existing benchmarks with proper redundancy [1]*
>
> To the best of our knowledge, no benchmark exists for our task. The task referred to in [1] is to find the important properties (by ranking the properties) to *describe* a *single entity*, while our goal is to select the important properties whose values enable us to *compare* *several entities*. We think that these two tasks are therefore very different. We also notice that the benchmark in [1] is limited to persons, which is only a small part of what we consider in Wikidata.
>
> *W3: The evaluation is also limited as it lacks baselines*
>
> There are 3 baselines corresponding to the approaches identified as the closest to ours in the related work: Facet [Oren et al., 2006], All [Petrova et al., 2017] and Infobox (using the templates of Wikipedia).
>
> *W3: the proposed contextual reference levels seem to be a generalization of the peer groups used in Recoin [1,2], and as such the paper should compare with that simple, class-based baseline*
>
> We have some difficulties making the link with the references [1,2] (especially since the targeted properties are different: interesting missing property vs interesting comparison feature). In this work as in ours, the goal is to determine whether a property is interesting by relying on other similar entities. But the methods are very different because they do not handle the same data and in the same way: [1] uses, for instance, full text by relying on topic modeling (LSI and LDA) and in [2], the similarity score benefits from the frequency of properties while our interestingness measure relies on the frequency of the objects for a same property.

---

> > ### Comment · AnonReviewer4 · 2021-01-31
> > **Rebuttal clarifies most of my concerns, have upgraded my score**
> >
> > Thanks for the response, which clarifies most of my concerns.
> >
> > The only point I still see doubt is the relation to Recoin - whether its goal and features are similar or not, it is an open hypothesis to disprove that those features, and/or goal, make its super simple property counting approach not suited for comparison table generation. In the same vein, the paper would benefit from experimentally proving that property rankings for single entities are not suited for pairs (e.g., are the attributes for the comparison tables "Ronaldo-Biden" and "Ronaldo-Turing" really that different?
> >
> > I hope the authors will include their clarifications also in the paper. Since my other concerns have been clarified, I have upgraded my recommendation.

---

### Official Review · AnonReviewer5 · 2021-01-06
**Comparison Table Generation from Knowledge Bases**

**Rating:** 2
**Confidence:** 3
**Impact:** 4
**Design And Technical Quality:** 5

**Review:**

The authors propose a method to automatically create comparison tables between entities from a Knowledge Base. This method creates such comparison tables automatically, only given the entities and a Knowledge Base as input.

The paper is very well written and addresses an interesting problem. The research problem, the approach, implementation, and evaluation are clearly described and show the effectiveness and efficiency of the proposed method.

*Related Work*

The authors claim that their method is the first to solve fully automatic comparison table generation and distinguish it from several related approaches such as infobox template generation or faceted search.

*Method*

The paper describes an approach to solve two main challenges: (1) the definition of an interestingness measure and (2) the efficient evaluation of such a measure.

The proposed interestingness measure as well as an algorithm to determine it are well defined and clearly described. Based on these definitions, upper and lower bounds for the measure are presented which enable early stopping in its calculation and hence allow for efficient evaluation.

*Experiments*

To evaluate the proposed method, the authors present a benchmark dataset that they created specifically for this task. With this dataset, they show that their method performs better than two baselines and a selected facet generation method. It is further demonstrated that the efficient evaluation of the interestingness measure reduces the time required to generate the comparison tables by a significant amount.

*Minor Remarks*

- the font of the last paragraph on page 4 seems to be incorrectly formatted
- Sections 3 and 4 are written very well and easy to understand. Except Definition 4 in Section 4.1. A short verbal explanation of Definition 4 and Property 1 would make this part easier to understand.


**Anonymity:**

Yes, I would like my review to remain anonymous.

**Reuse And Availability:**

2: Low

**Strong Points:**

- very well written
- clear definition of problem and proposed solution
- good description of experimental evaluation

**Subreviewer:**

I submitted this review.

**Weak Points:**

- benchmark dataset only accessible with password, this limits reproducibility

---

> ### Author Rebuttal · Authors · 2021-01-29
>
> Thank you for the comments. In case of acceptance, more verbal explanations will be added for Definition 4 and Property 1, and the download section (including the benchmark) is now publicly available without password.

---

### Official Review · AnonReviewer1 · 2021-01-12
**A new task in KB summary**

**Rating:** 1
**Confidence:** 4
**Impact:** 2
**Design And Technical Quality:** 3

**Review:**

This paper presents a new task in knowledge base summary, namely the generation of **comparison tables** between a set of entities from a given KB, for the purpose of their comparison along a number of dimensions (relations/features).

Besides the proposal of this new task, the main contributions of the paper are the following:

* A measure of *interestingness* for features that determines whether a feature should be included in a comparison table.

* An efficient method of generating fewer SPARQL queries to evaluate interestingness.

Detailed comments.

* The notion of interestingness is defined in terms of "contextual reference level" of features, and it is calculated from the statistics of the background knowledge base. However, the key definition, Definition 3, seems to contain some errors. At the top of page 6, in the definition of $crl_C(f, E, \mathcal{K})$, it is defined as "the probability of observing the values $f(e_i)$ of at least one entity $e_i$". However, the definition is defined as the "probability" of logical disjunction of set intersections being non-empty, which is a 0 or 1 value, but not a value in [0, 1]. This is wrong. Instead, $crl_C$ probably should have been specified as the *proportion* of common entities in each pair of $f(e_i)$ and $f(s_i)$.

* Similar errors appear later in the paper, in Definition 4 and Property 1 for example. These probability definitions are key to the proposed algorithm, and these errors made it a bit hard to understand it.

* Can you explain, in Table 1, why $crl$ of "spoken language" is 0.472? If I understood the definition of context correctly, the context is the set of all entities that speak English except these two entities. So, does 0.472 mean each person in the context speaks roughly 2 languages?

* In Algorithm 2, line 2, why are the entities sorted in the ascending order? If the goal is to minimise the size of the context, wouldn't it make more sense to try to eliminate the largest context first?

* It is not clear how the processing steps and the resulting dataset described in Sec 6.1 is related to the subsequent subsections. Specifically, on which dataset was the evaluation in Sec. 6.2 performed? How were the ground-truth labels obtained from the 6 human annotators?

* What do you mean by "Only 80 evaluations were common"?

* Besides precision and recall, it makes sense to report F1 as well.

* Even though this is a new task, maybe methods from other related tasks can be applied here. For example, entity summarisation methods aim at producing a subgraph about a given entity. A comparison table could be extracted from subgraphs of sets of entities $E$.

## Rebuttal
Thanks for your response, but I'm still confused by the definition of $crl_C$ on top of page 6. Specifically, it is defined as the _logic disjunction of Boolean formulas_ stating whether each set intersection is empty or not. In other words, it is a formula of $F_1\lor\ldots\lor F_n$, where each $F_i$ is of the form $E_1\cap E_2\neq\emptyset$. $F_1\lor\ldots\lor F_n$ takes on a 0 or 1 value, so how can you derive a probability from it?

I feel the clarity of the paper needs to be improved. Therefore I stand by my original scores.

**Anonymity:**

Yes, I would like my review to remain anonymous.

**Reuse And Availability:**

4: High

**Strong Points:**

* An interesting new task.

* The paper is in general well written.

* A statistics-based method that outperforms compared existing methods on both accuracy and efficiency.

**Subreviewer:**

I submitted this review.

**Weak Points:**

* Some errors in the description of the methodology, making it hard to comprehend.

* The evaluation setup was not described clearly.

* Some other baseline methods (e.g. entity summarisation) could have been compared.

---

> ### Author Rebuttal · Authors · 2021-01-29
>
> Thanks a lot for all the detailed comments and corrections. We are going to review related work about the KB summarization methods that are entity-centric; could you recommend us some papers?
>
> Regarding the definition of $crl_C$ (top of page 6), we are not sure we fully understand your proposed correction. Indeed, we rather look at the proportion of entities in $C$ (denoted by $s_i$) having at least one value in common with those of an entity in $E$ (denoted by $e_i$). Therefore, $0.472$ does not mean that each person in the context speaks roughly 2 languages (especially since there is a combination of several measures $crl_{C_i}$ corresponding to each context $C_i$). An approximate explanation would be: $0.47$ is the probability of having someone who speaks English in at least one common context with Ada Lovelace and Alan Turing.
>
> In Algorithm 2 (line 2), the entities are sorted in the ascending order because the intersection between sets is reduced faster when these sets are small, allowing fewer sets to be selected.
>
> Concerning the ground truth, there are 1,275 evaluations covering 1,195 features among 11,853 candidate features coming from 1,000 comparison tables. This random coverage of 10% of the candidate features is sufficient for statistical validity. Regarding the Kappa coefficient, there are 74 agreements (24  $yes$/50 $no$) and 6 disagreements (3 $yes/no$ + 3 $no/yes$) leading to $P_{yes}=0.113$, $P_{no}=0.438$ and Kappa=((74/80) - ($P_{yes}$+$P_{no}$)) / (1 - ($P_{yes}$+$P_{no}$))=0.832.
>
> Due to lack of space, we have not shown the F-measure that can be deduced from precision and recall. We preferred to show the accuracy, which takes into account true negatives ignored by precision and recall.

---

> > ### Comment · AnonReviewer1 · 2021-02-01
> > **I'm still confused by the definition...**
> >
> > Thanks for your response, but I'm still confused by the definition of $crl_C$ on top of page 6. Specifically, it is defined as the _logic disjunction of Boolean formulas_ stating whether each set intersection is empty or not. In other words, it is a formula of $F_1\lor\ldots\lor F_n$, where each $F_i$ is of the form $E_1\cap E_2\neq\emptyset$. $F_1\lor\ldots\lor F_n$ takes on a 0 or 1 value, so how can you derive a probability from it?
> >
> > I feel the clarity of the paper needs to be improved. Therefore I stand by my original scores.

---

### Official Review · AnonReviewer2 · 2021-01-13
**Solid formal description of an interesting approach**

**Rating:** 2
**Confidence:** 3
**Impact:** 3
**Design And Technical Quality:** 4

**Review:**

The authors discuss the problem of comparison table generation, i.e. generating a table of facts for a set of entities to aid their comparison. To that end, they present VERSUS, an approach to generate such comparison tables by exploiting knowledge about shared contexts of the entities to be compared. To evaluate whether a feature is useful for the comparsion of entities, they compute its contextual reference level (crl). The crl of a feature is high, if it has a high probability of occurring in the context of the compared entities (i.e., in entities that are connected to the compared entities by a certain relation). With a dataset (CFB) that is part of their contribution, the authors compute a lower bound for the crl of acceptable features and evaluate the performance of VERSUS for this novel problem.

All in all, the paper is well written and has a strong formal description of the problem and the approach to solve it. At the same time, sufficient examples are provided to aid the understanding.

The only minor doubt that I have is about the created benchmark. The authors take only the entities with the highest degree of incoming facts. Here, I am unsure, whether this distorts the significance of the created benchmark as it only deals with very prominent examples. This has as a consequence that the evaluation results are at least not representative for the average comparison of two entities.

However, as I do not have any major points of concern about this paper, I recommend to accept it.

AFTER REBUTTAL
I acknowledge the answers of the reviewers and keep my score.

**Anonymity:**

Yes, I would like my review to remain anonymous.

**Reuse And Availability:**

4: High

**Strong Points:**

- Straight-forward approach that seems to be working quite well
- Very strong formal description of problem and approach, but also sufficient examples to aid understanding

**Subreviewer:**

I submitted this review.

---

> ### Author Rebuttal · Authors · 2021-01-29
>
> Thank you for the comments. Concerning the representativeness of the benchmark, we have selected the two entities with the highest degree of incoming facts but among classes having only few entities sometimes. For this reason, the benchmark also contains less popular pairs of entities.

---

### Official Review · AnonReviewer3 · 2021-01-14
**Comparison Table Generation from Knowledge Bases.**

**Rating:** 2
**Confidence:** 4
**Impact:** 3
**Design And Technical Quality:** 4

**Review:**

The authors propose an approach named VERSUS that aims to help a user to compare a set of entities described in RDF thanks to comparison tables. These tables represent a set of properties that can be selected thanks to an interestingness measure, called the Contextual Reference Level (Crl), that expresses that a property value of one of the entity is rather frequent in a context limited to entities that share some property values with the compared entities. The main properties of this measure are given and an efficient algorithm has been defined to select the contexts and the relevant properties in which the needed sparql queries can be minimized thanks to the estimation of an upper bound and a lower bound of the crl value.
The approach has been evaluated on almost 1 200 properties that describe ‘popular’ couples of Wikidata entities (entities that frequently instanciate the range of a property) that belongs to 1000 different classes. The results are compared to two other methods (properties that appear in a Wikipedia infobox and automatically constructed facets).

The approach is interesting since tables are easy to visualize, interpret and manipulate. The proposed approach is not defined for a specific domain or application and doesn’t require any manually specified knowledge. The interestingness measure is original and based on a rather simple notion of context that is not based on ontology semantics, spatial or temporal information, user or application. All the properties such that their crl values are greater than a given threshold can be selected using a sparql endpoint with an algorithm that is much more faster than a naïve one. The experimental evaluation shows that the precision is rather high on the CFB gold-standard constructed thanks to 6 evaluators and that recall or precision is higher than those that can be obtained when the results are compared to two existing approaches. The constructed benchmark (CFB)  is publicly available.
Furthermore, the approach is well argued and formalized.

However I have some remarks:

-Even if your approach is not dedicated to some application, it would be more convincing if you better justify the approach by presenting in the introduction examples of applications that can exploit such comparison tables .

- The approach do not allow to take into account property paths but this is one of the main difficulty when a user have to compare RDF descriptions. Furthermore, how can an evaluator decide that a property is similar or not , when the URI of an object property is shown (depending on the KG, the URI is not always meaningful) ? You also have to say why you have only selected object properties in the experimentation.

-The fact that the computation of the crl values can be stopped do not allow to rank the selected property according to the crl value. However, a ranking could make it possible to select more properties if necessary. Indeed, the evaluation shows that the number of selected properties can be rather small (3 properties in average when the threshold is fixed to 0.1, even if you have selected “well-known” entities). Using this approach, the crl value could help to select the n best ranked property whatever the crl value is.

- I do not understand what is said about the property 3 (p7) : It is said that “This property underlines that the value of contextual reference level is always underestimated when some facts are missing, i.e. if new facts are added in the
knowledge base, then the contextual reference level of a feature can only increase.” Property 3 says that if the number of values of a property increases in E the crl value can only increase. However, if some new instances and property values appear in the context, crl can only be equal or decrease. So why do you conclude that a property remain relevant when some facts are added to the KG ? (it is only valid when for the RDF descriptions in E ? no ?).

- The evaluation does not show the impact of considering larger sets for E.

Minor remarks :
- Why don’t you consider that relevant properties can belong to r-1 ? (sometimes it is just a matter of modelization for object properties).

- You have to say why you have chosen to compare this approach to the automatic facet extraction methods described in [8] and not another approach based on attributes whose values are frequently observed since [8] (such as [6], even if 6 only deal with infobox properties).

- Add the way the accuracy has been computed, since there is 17.5% of “don’t know” in the gold-standard.

- I really think that selecting properties even if there are null values can be interesting. Have the evaluators also chosen properties for which one null value appears ? It would be interesting to add this information in the paper.

- It would also be very informative to show how many entities have no selected properties at all, for your approach and the facet ones (for sure it is not the case with the infoboxes).

-  I think that it can be a bias to ask to the evaluator  to select properties when other properties are not shown for an entity (e.g. in DBPedia, some locations are expressed using a multivalued property, and each value represents a part of the information). Was it evaluated one property by one property ? (it is not so clear for me).

- Since the recall is not so high, while the precision remains high when the threshold is fixed 0.001, it would be informative to see how precision and recall evolve for some lower thresholds.

-I think that it can be also relevant to mention briefly some of the methods developed for RDF graph summarization since some of these approaches aim at extracting the most interesting triples to be shown to a user about a subject.


**Anonymity:**

Yes, I would like my review to remain anonymous.

**Reuse And Availability:**

4: High

**Strong Points:**

-A well-argued and well-formalized approach that can help a user to compare entities of a KG without considering additional knowledge.
-An original interestingness measure for the properties.
-An evaluation that shows good precision and higher recall or precision when compared to 2 other approaches and a baseline.
-Two efficient algorithms to select contexts and properties.
- The tool and the CFB gold-standard are available.

REBUTTAL :
Thank you for the clarity of your answers about the technical points mentioned in my minor remarks (property 3, accuracy) and about the very pragmatic (but tested)  answer about the exploitation of r-1.
I think that a sentence to explain why datatype properties are not considered can be easily added.
Furthermore, even if I still believe that for some pairs of individuals it would be more interesting to consider the top-K properties using a "less efficient" approach, I am convinced by your answers. I think that the algorithm and the interestingness measure are both of interest, even if the approach can be extended/generalized by considering different research directions.

**Subreviewer:**

I submitted this review.

**Weak Points:**

-	An approach based on a very strong hypothesis that states that the relevant properties can be defined without considering a specific task/application. So, the impact of using this tool  is not so easy to estimate.
-	The feature selection only consider property paths of length 1 in the RDF descriptions.
-	An evaluation that is “only” conducted on an extract of Wikidata. Some experimental conditions/choices and some qualitative evaluation analysis are missing.
-	An obtained recall that is not so high.

---

> ### Author Rebuttal · Authors · 2021-01-29
>
> Thank you very much for your various feedback.
>
> Here are some details/remarks:
> - Concerning the justification of the method, it would indeed be interesting to specify the usefulness of the comparison tables for compensatory decision making.
> - Property 3 says that if the number of values of a property increases for any entity (not only the compared entities E) the crl value can only increase. This is correct because we assume that the context \cal C is the same on both sides of the inequality.
> - Concerning the choice of [8], we think that [6] (based on infobox) would have a weak recall because this recall is inferior to that of infobox baseline (see Fig. 1). Besides, the metric proposed in [5] is conceptually very close to that proposed in [8].
> - The evaluations denoted by “don’t know” are ignored in the calculation of accuracy.
>
> Clearly, the method can be improved on several points that you raised:
> - We are aware that it is difficult to consider paths rather than using RDF properties directly.
> - For other KGs, the evaluation interface might be modified for making the generated comparison table understandable.
> - We only selected object properties because our approach is not suitable for (quasi-)continuous literal types (like numerical values, timestamps).
> - Indeed, our method does not allow to rank the selected property according to the crl value. To be able to select the top-k properties, it would be necessary to add a branch-and- bound process, which after each property refines the minimum crl threshold to retain only the k best ones.
>
> We made some choices due to lack of space:
> - There is no experiment neither on another KB, nor on more than 2 entities. But, for this rebuttal phase, we add examples as illustration in a new version of the online supplementary material corresponding to examples (that are already given in the source code of the prototype).
> - We have carried out experiments on the properties r^-1 (implemented in the source code). There are a lot less comparison features extracted with the inverse properties. In addition, their functionality (number of facts per entity) is sometimes very high making the comparison tables less readable.

---

### Decision · Program_Chairs · 2021-02-23

**Decision:**

Accept

**Comment:**


The reviewers agreed that this work presents major strengths. In particular, the reviewers welcome the novelty of the addressed problem and the soundness of the proposed solution VERSUS. While some reviewers considered that VERSUS was not properly evaluated with existing related approaches, this work lays the foundations for further analyses and more sophisticated solutions to the problem of computing comparison tables. Based on the evaluations of all the reviewers, this paper is recommended for acceptance.

In preparation for the camera-ready, we kindly ask the authors to address the comments raised by the reviewers and include the requested clarification. In particular, the readability of the paper should be improved in two points:

1. Definition 4, including the formulations at the beginning of page 6 (which are correct but hard to follow),

2. Property 1. The reviewers consider that the intuition of these definitions should be introduced more clearly as they are central to the rest of the approach.